# Autoformalizing Mathematical Statements by Symbolic Equivalence and Semantic Consistency

**Zenan Li**[1*]   **Yifan Wu**[2*]   **Zhaoyu Li**[3]   **Xinming Wei**[2]
**Fan Yang**[4]   **Xian Zhang**[4]   **Xiaoxing Ma**[1]
[1]State Key Lab of Novel Software Technology, Nanjing University, China
[2]Peking University, [3]University of Toronto
[4]Microsoft Research Asia
lizn@smail.nju.edu.cn, yifan.wu@stu.pku.edu.cn,
zhxian@microsoft.com, xxm@nju.edu.cn

## Abstract

Autoformalization, the task of automatically translating natural language descriptions into a formal language, poses a significant challenge across various domains, especially in mathematics. Recent advancements in large language models (LLMs) have unveiled their promising capabilities to formalize even competition-level math problems. However, we observe a considerable discrepancy between pass@1 and pass@$k$ accuracies in LLM-generated formalizations. To address this gap, we introduce a novel framework that scores and selects the best result from $k$ autoformalization candidates based on two complementary self-consistency methods: *symbolic equivalence* and *semantic consistency*. Elaborately, symbolic equivalence identifies the logical homogeneity among autoformalization candidates using automated theorem provers, and semantic consistency evaluates the preservation of the original meaning by informalizing the candidates and computing the similarity between the embeddings of the original and informalized texts. Our extensive experiments on the MATH and miniF2F datasets demonstrate that our approach significantly enhances autoformalization accuracy, achieving up to 0.22-1.35x relative improvements across various LLMs and baseline methods. The data and code are available at `https://github.com/Miracle-Messi/Isa-AutoFormal`.

## 1   Introduction

Autoformalization is the automated process of translating from natural language expressions into a formal language [1–4]. Successful autoformalization can alleviate the demand for extensive human expertise and reduce the substantial manual formalization efforts [5–8], as well as fundamentally bridging the gap between natural (or so-called "informal") and formal languages [9], which potentially catalyzes breakthroughs in many fields such as mathematical theorem proving, software/hardware verification, and autonomous planning [10–14]. Despite decades of research, the practical application of autoformalization remains limited because traditional methods often necessitate either predefined domain-specific languages or hard-coded translation rules [15–18].

Recently, large language models (LLMs) have shown promising performance in autoformalization, especially in formalizing mathematical statements [19–21]. For instance, using Codex [22] with few shot examples can achieve a 25.3% success rate in formalizing high-school level problems from the MATH [23] dataset. Nevertheless, the autoformalization capability of LLMs has not been fully exploited. As shown in Figure 1, even advanced LLMs like GPT-4 [24] struggle with translating a

---

*Equal contribution. This work was partially done during Zenan's and Yifan's internships at MSRA.

> **Prompt:**
> Natural language version: "Find the product of $0.\overline{6}$ and 6. The answer is 4."
> Translate the natural language version to an Isabelle version.
>
> | **GPT-4 output (No.1):** | **GPT-4 output (No.2):** | **GPT-4 output (No.3):** | **GPT-4 output (No.4):** |
> |---|---|---|---|
> | theorem | theorem | theorem | theorem |
> | fixes x y :: real | fixes a b :: real | fixes x y :: real | fixes x y :: real |
> | assumes "x = 0.66" | assumes "a = 2/3" | assumes "x = 2/3" | assumes "x = 0.6666" |
> | and "y = 6" | and "b = 6" | and "y = 6" | and "y = 6" |
> | shows "x * y = 4" | shows "a * b = 4" | shows "x * y = 4" | shows "x * y = 4" |

Figure 1: An illustrative example of autoformlization. The mathematical statement from the MATH dataset is translated into a formal version by GPT-4. Only two formalization results (No.2 and No.3) are correct, while the others fail in the grounding ($0.\overline{6} \rightarrow 2/3$).

seemingly straightforward statement (i.e., calculating $0.\dot{6} \times 6$), due to its unreliability of grounding a recurring decimal to its exact fractional equivalent ($0.\dot{6} \rightarrow 2/3$).

Although the initial attempt at formalizing the previous example fails, we observe that multiple generations of GPT-4 can often successfully cover the correct formalization. In other words, a significant disparity exists between pass@1 (the top-1 generation is correct) and pass@$k$ (one of the top-$k$ generations is correct) in autoformalization by LLMs. We further confirm this phenomenon by checking the autoformalization accuracy on the MATH and miniF2F [25] datasets. The curves depicted in Figure 2 demonstrate that pass@$k$ accuracy can be consistently improved with additional generations, resulting in an accuracy gap ranging from 19.5% to 26.5% between pass@1 and pass@10.

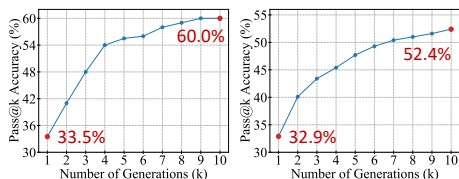

To bridge the performance gap between pass@1 and pass@$k$, a natural approach is employing the idea of self-consistency [26, 27] to rank the $k$ autoformalization candidates and then selecting the most consistent one. However, compared to the standard self-consistency techniques used in mathematical reasoning and code generation with LLMs, applying this method faces unique challenges. In mathematical reasoning, self-consistency is always derived by comparing the final answers from different generations,

Figure 2: Pass@$k$ curves for GPT-4 autoformalization on the MATH (left) and miniF2F (right) datasets. The results show that LLMs can achieve higher coverage of correct formal statements with an increasing number of generated candidates up to $k = 10$. Beyond this point, the improvement gradually diminishes as $k$ continues to increase.

yet this approach struggles with inconsistencies in symbolic variable declarations ($(\mathtt{a}, \mathtt{b})$ vs $(\mathtt{x}, \mathtt{y})$). In code generation, while self-consistency relies on comparing execution behaviors across different generations, this approach is less viable for formalized mathematical statements, which lack the necessary test cases for such evaluations.

To address these challenges, we propose a novel framework that establishes the self-consistency of autoformalization from two innovative and complementary dimensions: *symbolic equivalence* and *semantic consistency*. Symbolic equivalence generalizes traditional comparisons like final answers and execution behaviors to verify the logical equivalence among autoformalization candidates. This is achieved by using automated theorem provers such as Sledgehammer [28], Z3 [29], and CVC5 [30]. On the other hand, semantic consistency rectifies unintended reasoning discrepancies that symbolic equivalence might overlook by measuring the embedding similarity between the re-informalized (back-translated [31]) result and the original natural language statement. This comparison helps to ensure that the autoformalization process preserves the intended meaning and coherence of the original statement. To harness the strengths of both consistency methods, we also develop three strategies to combine the scores from these approaches.

We conduct extensive evaluations on two widely used mathematical datasets, MATH and miniF2F, to validate the efficacy of our proposed methods. The experimental results demonstrate that symbolic equivalence and semantic consistency are synergistic, and our combination strategy can achieve final improvements up to 7.8% and 10.7% using GPT-4 compared with the baseline approaches. The relative efficiency of our method, ranging from 8.4% to 21.9%, indicates that our approach

can significantly reduce the manual effort required for verifying or labeling formalization results, efficiently minimizing human intervention in correcting and validating outputs. Additionally, we extend our experiments to five proprietary or open-source LLMs, showing the consistent effectiveness of the proposed methods.

In summary, this paper makes the following main contributions: (1) identifying the performance gap between pass@1 and pass@$k$ for LLMs in autoformalization tasks; (2) introducing two self-consistency methods, symbolic equivalence and semantic consistency, and three combination strategies to enhance LLM autoformalization performance; (3) providing extensive experiments across various model sizes on two popular datasets, confirming the efficacy of the proposed approach.

## 2 Background and Related Work

**Formal mathematics.** Formal mathematics aims to establish a rigorous framework to express mathematical theorems and proofs in a format that can be verified by a computer through the application of logical rules. Interactive theorem provers, such as Isabelle/HOL [32], Coq [33], and Lean [34], provide environments for encoding and verifying mathematical proofs programmatically. For decades, researchers have used these tools to manually formalize a range of challenging mathematical concepts and theorems [7, 8, 35–37]. However, translating mathematics into a language that theorem provers can interpret often requires a deep understanding of both the mathematics involved and the syntax of the target formal language. Therefore, the formalization process is always labor-intensive even for large groups of experts, creating a significant bottleneck in this field.

**Autoformalization with LLMs.** To mitigate the laborious process of manual formalization, recent advances have explored the potential of LLMs in autoformalization [38]. A stream of research focuses on autoformalizing mathematical statements [19–21, 39–42]. For instance, FIMO [41] employs GPT-4 with reflection to formalize problems from the International Mathematical Olympiad. ProofGPT [20] and MMA [21] train LLMs on large-scale datasets with both informal and formal mathematical data to evaluate their performance on statement autoformalization. Concurrently, another research direction investigates the autoformalization of mathematical proofs [10, 11, 43–48]. For example, DSP [10] utilizes LLMs to draft informal proofs and map them into formal sketches, with automated theorem provers employed to fill in the missing details in the proof sketch. Besides these efforts, several studies [19–21] explore the performance of LLMs for the inverse process of formalization, i.e., informalization, which translates formal statements back into natural language.

**Self-consistency for LLMs.** Self-consistency was originally proposed to boost the mathematical reasoning capability of LLMs [26, 49–52]. This approach aims to identify the homogeneity among multiple generations, thereby bridging the performance disparity between pass@1 and pass@$k$. In contrast to other techniques, such as training an additional verifier/re-ranker [53, 54], or directly fine-tuning the model [55], self-consistency is entirely data-free, making it readily implementable with off-the-shelf LLMs without incurring the so-called "alignment tax" associated with additional computational costs [55]. Recently, self-consistency has been further adapted to code generation, which closely resembles autoformalization since they both involve formalizing natural language statements. However, in code generation, self-consistency for LLMs typically relies on the execution information from test cases [27, 56–58], e.g., whether the two programs produce the same output for identical test inputs. Therefore, this strategy is not applicable to autoformalization due to the absence of test cases for mathematical statements.

## 3 Methodology

Our framework, as illustrated in Figure 3, comprises four steps to enhance the autoformalization process of LLMs. Initially, LLMs generate $k$ autoformalization candidates for a given mathematical statement in natural language. Subsequently, our framework establishes the symbolic equivalence among these candidates and assigns a symbolic score to each based on the derived equivalence classes. Each formal statement is then re-informalized using LLMs, and the semantic score is computed by comparing the embeddings of the re-informalized text and the original statement. Finally, our framework normalizes and combines these scores to rank the autoformalization candidates and determine the final formalization results.

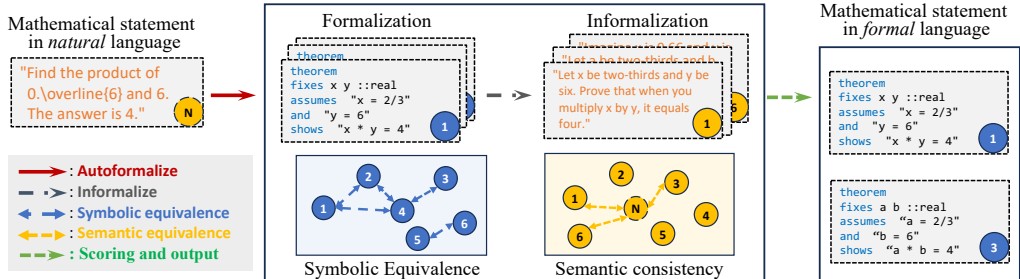

Figure 3: The overview of our autoformalization framework. In the framework, symbolic equivalence is constructed among formalized statements, and semantic consistency is computed between the informalized statements and the original statement. The scores from these two evaluations are combined to rank and select the final formalization results.

## 3.1 Symbolic Equivalence

We first instantiate self-consistency as symbolic equivalence among autoformalization candidates. The rationale behind the symbolic equivalence is straightforward: *correct formalizations are logically equivalent, even when expressed with varied symbols.* To establish symbolic equivalence, we decompose formal statements into their premises and conclusions. Symbolic equivalence between two statements is then defined by the logical equivalence of both their premises and conclusions.

The formal definition of symbolic equivalence is presented in the following. Within this definition, we further assume that the premises are not intrinsically contradictory, ensuring that the two involved mathematical statements are well-defined.

**Definition 1 (Symbolic equivalence)** *Let two mathematical statements $\Psi_1$ and $\Psi_2$ in formal language be expressed as $\mathcal{P}_1 \rightarrow \mathcal{Q}_1$ and $\mathcal{P}_2 \rightarrow \mathcal{Q}_2$, and suppose their premises $\mathcal{P}_1$ and $\mathcal{P}_2$ are not tautologies. Then, the two statements are called symbolically equivalent if the two logical equivalences, i.e., $\mathcal{P}_1 \equiv \mathcal{P}_2$ and $\mathcal{Q}_1 \equiv \mathcal{Q}_2$, both hold.*

The two logical equivalences induced by the symbolic equivalence can be determined through existing automated theorem provers (ATPs). Additionally, the validity of the premises can also be checked by replacing the conclusion $\mathcal{Q}$ with a contradictory result (e.g., $0 = 1$) and verifying $\mathcal{P} \rightarrow \mathcal{Q}$. If this vacuous form can be proved, then the corresponding premise $\mathcal{P}$ is a contradiction.

It is also worth noting that variable misalignment between two statements remains a challenge for validating symbolic equivalence using ATPs. For instance, the examples in Figure 1 (No.2 and No.3) can not be proved symbolically equivalent due to the inconsistent variable declarations. Therefore, we should perform variable matching beforehand, ensuring that the symbolic equivalence can be well recognized even in these cases.

Nevertheless, exhaustively checking all possible variable mappings is always impractical due to the combinatorial explosion. For two statements expressed as $\mathcal{P}_1(x_1, \ldots, x_n) \rightarrow \mathcal{Q}_1(x_1, \ldots, x_n)$ and $\mathcal{P}_2(y_1, \ldots, y_n) \rightarrow \mathcal{Q}_2(y_1, \ldots, y_n)$, each has $n$ variables, there are $n!$ possible bijective mappings to be checked, which is excessively time-consuming when $n$ is large. To address this issue, we propose to standardize the formal statement $\mathcal{P}(x_1, \ldots, x_n) \rightarrow \mathcal{Q}(x_1, \ldots, x_n)$ by the following two cases:

(1) If the conclusion is in the form of a numerical relation, i.e., $\mathcal{Q}(x_1, \ldots, x_n) := f(x_1, \ldots, x_n) \bowtie 0$, where $f$ represents any function and $\bowtie \in \{\leq, \geq, <, >, =, \neq\}$, we introduce a new variable $\alpha$ and derive the standard format $\tilde{\mathcal{P}}(\alpha; x_1, \ldots, x_n) \rightarrow \tilde{\mathcal{Q}}(\alpha)$ with

$$\tilde{\mathcal{P}}(\alpha; x_1, \ldots, x_n) := \mathcal{P}(x_1, \ldots, x_n) \wedge (\alpha = f(x_1, \ldots, x_n)), \quad \tilde{\mathcal{Q}}(\alpha) := \alpha \bowtie 0.$$

The two statements are reduced to $\tilde{\mathcal{P}}_1(\alpha; x_1, \ldots, x_n) \rightarrow \tilde{\mathcal{Q}}_1(\alpha)$ and $\tilde{\mathcal{P}}_2(\alpha; y_1, \ldots, y_n) \rightarrow \tilde{\mathcal{Q}}_2(\alpha)$, and thus the logical equivalences $\tilde{\mathcal{P}}_1(\alpha) \equiv \tilde{\mathcal{P}}_2(\alpha)$ and $\tilde{\mathcal{Q}}_1(\alpha) \equiv \tilde{\mathcal{Q}}_2(\alpha)$ can be checked through leaving $x_1, \ldots, x_n$ and $y_1, \ldots, y_n$ as auxiliary variables.

(2) For non-numerical cases (e.g., $\mathcal{Q}(x) := \texttt{is\_even}(x)$), we have to conduct a variable alignment. Instead of enumerating all variable mappings, we view the variables in each statement as a set of graph vertices, and thus the variable alignment is transformed into a bipartite matching task (where

$(x_1, \ldots, x_n)$ and $(y_1, \ldots, y_n)$ form two disjoint and independent vertex sets) [59, 60]. Furthermore, we simply set the edge weight of the graph by the string edit distance [61, 62], and only partially enumerate variable mappings corresponding top-$k$ maximum bipartite matching.

To further clarify, we provide an example for each case in Appendix C. For evaluating symbolic equivalence, we assign the symbolic score to each formalization using the proportion of its corresponding equivalence class, which is also commonly used in mathematical reasoning and code generation.

## 3.2 Semantic Consistency

Next, the self-consistency is instantiated as the semantic consistency between the formalization and its corresponding informal version. The rationale of the semantic consistency is also clear: *An autoformalization result is accurate if it can be re-informalized to a version consistent with the original statement in natural language.* By introducing the embedding similarity [63, 64] to measure the consistency between the original text and the twice-processed (autoformalized then informalized) version, we can define the $\tau$-semantic consistency as follows.

**Definition 2 (Semantic consistency)** *Let the original mathematical statement in natural language and its formalization candidate be $\Phi$ and $\Psi$, and suppose that $\Psi$ is further informalized into a new natural language statement $\tilde{\Psi}$. Then, the formal statement $\Psi$ is $\tau$-semantically consistent with the original statement $\Phi$ if the embedding similarity between $\tilde{\Psi}$ and $\Phi$ satisfies $\mathrm{Sim}(\tilde{\Psi}, \Phi) \geq \tau$.*

Semantic consistency primarily measures the error incurred in both the formalization and informalization processes. However, in most cases, it can be approximately reduced to measuring the error in autoformalization. This is because informalization is much easier and accurate than formalization [21]. For instance, considering the formalization of the statement "Determine the range of $e^2$", LLMs should inference the type of the exponential $e$, determining whether to use `powr` or `^` as the grounding of 'power'. On the contrary, these two expressions are both translated back into the term 'power' during informalization.

Compared to symbolic equivalence, semantic consistency can avoid the *unintended reasoning* problem. Elaborately, continuing the example in Figure 1, the correct formalization "$(x = 2/3 \wedge y = 6) \rightarrow x * y = 4$" is identified as symbolically equivalent to the formalization "$4 = 4$", while the latter is trivial and unexpected. However, the difference between these formalizations can be successfully recognized by semantic consistency since the latter ruins the semantics in the informal statement.

Following existing machine translation techniques [65–67], we employ the BERT model [68] to generate embeddings for the informal statements. These embeddings are then compared using cosine similarity to evaluate semantic consistency.

## 3.3 Combination of Two Scores

Given $k$ autoformalization candidates, and denote their scores of symbolic equivalence and semantic consistency by $s_1^{\mathrm{sym}}, \ldots, s_k^{\mathrm{sym}}$ and $s_1^{\mathrm{sem}}, \ldots, s_k^{\mathrm{sem}}$, respectively. We first normalize them using the softmax function, i.e., $\hat{s}_i^{\mathrm{sym}} = s_i^{\mathrm{sym}} / \sum_{j=1}^k s_j^{\mathrm{sym}}$ and $\hat{s}_i^{\mathrm{sem}} = s_i^{\mathrm{sem}} / \sum_{j=1}^k s_j^{\mathrm{sem}}$ for $i = 1, \ldots, k$. Then, we propose three strategies, i.e., log, linear, and quadratic, for the combination of two scores. In particular, the final score $\hat{s}_i$ of the $i$-th autoformalization candidate is computed by

$$\text{Log combination:} \quad \hat{s}_i = \alpha \log \hat{s}_i^{\mathrm{sym}} + (1 - \alpha) \log \hat{s}_i^{\mathrm{sem}},$$

$$\text{Linear combination:} \quad \hat{s}_i = \alpha \hat{s}_i^{\mathrm{sym}} + (1 - \alpha) \hat{s}_i^{\mathrm{sem}},$$

$$\text{Quadratic combination:} \quad \hat{s}_i = \alpha (\hat{s}_i^{\mathrm{sym}})^2 + (1 - \alpha)(\hat{s}_i^{\mathrm{sem}})^2,$$

where $\alpha \in [0, 1]$ is the hyperparameter controlling the trade-off between the symbolic equivalence and the semantic consistency, which practically can be tuned based on the validation set.

The overall procedure of our autoformalization framework is presented in Algorithm 1. The primary efficiency bottleneck of the algorithm lies in verifying symbolic equivalence. In the worst case, where no pair of autoformalization candidates are symbolically equivalent, and symbolic equivalence must be exhaustively validated $k(k - 1)/2$ times. However, many verifications of symbolic equivalence can be bypassed by leveraging the transitivity property of symbolic equivalence. Moreover, when requested to provide $n$ formalization results, we iteratively conduct the algorithm to rank the formal statements, with the selected result and formal statements in its equivalence class removed.

---
**Algorithm 1** Autoformalization based on symbolic equivalence and semantic consistency
---
**Input:** A mathematical statement in natural language $\Phi$,
      transform function $\rho$ (log, linear, or quadratic), hyperparameter $\alpha$;
**Output:** the autoformalization result with the highest score.
---
 1: Generate $k$ autoformalization candidates $\Psi_1, \ldots, \Psi_k$ by prompting LLMs;
 2: *Compute scores of the symbolic equivalence*
 3: **for** $i = 1, \ldots, k$ **do**
 4:    **for** $j = i+1, \ldots, k$ **do**
 5:       Standardize $\Psi_i$ and $\Psi_j$ into $\tilde{\Psi}_i := \tilde{\mathcal{P}}_i \to \tilde{\mathcal{Q}}_i$ and $\tilde{\Psi}_j := \tilde{\mathcal{P}}_j \to \tilde{\mathcal{Q}}_j$;
 6:       Check the logical equivalence $\tilde{\mathcal{P}}_i \equiv \tilde{\mathcal{P}}_j$ and $\tilde{\mathcal{Q}}_i \equiv \tilde{\mathcal{Q}}_j$ using ATPs;
 7:    **end for**
 8:    Compute $s_i^{\mathrm{sym}}$ as the size of the derived equivalence class;
 9: **end for**
10: *Compute scores of the semantic consistency*
11: **for** $i = 1, \ldots, k$ **do**
12:    Obtain the twice-processed version $\tilde{\Phi}_i$ by using LLMs to informalize $\Psi_i$;
13:    Compute $s_i^{\mathrm{sem}}$ as the cosine similarity between the embeddings of $\tilde{\Phi}_i$ and $\Phi$;
14: **end for**
15: *Combine the two scores*
16: Normalize $s_i^{\mathrm{sym}}, i = 1, \ldots, k$ and $s_i^{\mathrm{sem}}, i = 1, \ldots, k$ using the softmax function;
17: **for** $i = 1, \ldots, k$ **do**
18:    Compute the final score $s_i$ by $s_i = \alpha \rho(s_i^{\mathrm{sym}}) + (1 - \alpha)\rho(s_i^{\mathrm{sem}})$;
19: **end for**
---

## 4 Evaluation

In this section, we conduct a series of experiments to answer the following four research questions:

**RQ1: Efficacy** – Compared with baselines and alternatives, do our proposed methods (symbolic equivalence and semantic consistency) achieve better autoformalization performance?

**RQ2: Synergy** – Are symbolic equivalence and semantic consistency mutually complementary? Does the combination of them further boost the autoformalization performance?

**RQ3: Labeling-efficiency** – How much human effort in verifying or labeling the formalization results can be saved using our proposed methods?

**RQ4: Scalability** – Can our proposed methods be further enhanced by using stronger LLMs or ATPs?

### 4.1 Experimental Setup

**Dataset.** We evaluate the proposed methods on the MATH [23] and miniF2F [25] datasets, both of which encompass a wide range of mathematical problems designed for different levels of complexity and abstraction. The MATH dataset includes a variety of problem types, e.g., Algebra, Number Theory, Geometry, and so on. We randomly select a subset of 400 problems from the dataset to serve as our benchmark. The miniF2F dataset is specifically curated for evaluating LLM abilities in autoformalization and mathematical reasoning. It contains 488 Olympiad-level mathematical problems, each equipped with a formal statement as an oracle in Isabelle and Lean.

**Model.** We carry out the experiments on five proprietary and open-source models of varying parameter sizes, including Mistral-7B [69], Llemma-34B [45], DeepSeek-v2 [70], Codex (completion api) [22], and GPT-4 (version 0710) [71]. In addition, we employ few-shot prompting, and set the temperature of the generation process to 0.7 for all LLMs. The eight examples used, along with detailed prompts for autoformalization and informalization, are provided in Appendix F.

**Metric.** We use the unbiased $n@k$ accuracy (with $k$ generations) for performance evaluation, i.e., the percentage of problems for which the top-$n$ formalizations of $k$ generations can cover a correct version [22]. We apply different policies to determine autoformalization correctness on the MATH and miniF2F datasets, respectively. Specifically, the MATH dataset does not contain aligned formal

Table 1: Performance ($n@k$) of our methods (SymEq and SemCo) and comparison methods (Baseline, Naïve, and Cluster) on MATH and miniF2F datasets. The best performance of each $n$ is in bold. The results show that our proposed methods consistently achieves superior performance.

| METHODS | BASELINE | | | NAÏVE | | | CLUSTER | | | SYMEQ | | | SEMCO | | |
|---|---|---|---|---|---|---|---|---|---|---|---|---|---|---|---|
| $n$ | 1 | 2 | 3 | 1 | 2 | 3 | 1 | 2 | 3 | 1 | 2 | 3 | 1 | 2 | 3 |
| MATH | | | | | | | | | | | | | | | |
| Mistral-7B | 18.1 | 30.6 | 37.2 | 15.7 | 15.7 | 15.7 | 21.8 | 30.3 | 37.5 | **26.7** | **33.1** | 37.9 | 22.0 | 32.2 | **38.8** |
| Llemma-34B | 33.4 | 43.1 | 48.0 | 26.5 | 27.0 | 27.0 | 35.4 | 43.9 | 50.4 | **41.1** | **46.1** | **52.6** | 36.5 | **46.1** | 50.6 |
| DeepSeek-v2 | 37.3 | 42.4 | 44.4 | 24.7 | 26.6 | 27.0 | 34.9 | 41.7 | 45.6 | **42.4** | **44.9** | **46.9** | 38.6 | 43.6 | 45.8 |
| Codex | 43.8 | 48.7 | 52.4 | 22.4 | 22.9 | 23.1 | 42.6 | 48.0 | 51.2 | **46.1** | **50.8** | **54.3** | 44.9 | 49.6 | 52.9 |
| GPT-4 | 37.5 | 47.7 | 53.5 | 24.5 | 25.7 | 26.5 | 39.7 | 48.0 | 53.5 | **42.0** | **50.7** | **54.5** | 39.5 | 47.2 | **54.5** |
| MINIF2F | | | | | | | | | | | | | | | |
| Mistral-7B | 7.5 | 12.1 | 13.9 | 10.5 | 11.5 | 12.1 | 7.5 | 11.8 | 12.8 | **14.5** | **17.4** | **18.4** | 8.8 | 12.1 | 16.4 |
| Llemma-34B | 21.3 | 28.3 | 34.5 | 19.4 | 23.1 | 24.6 | 20.5 | 28.8 | 33.5 | **32.0** | **40.1** | **41.7** | 27.2 | 31.9 | 38.2 |
| DeepSeek-v2 | 26.8 | 29.8 | 31.7 | 28.0 | 31.3 | 32.9 | 27.6 | 30.1 | 30.9 | **28.4** | **31.7** | **33.8** | 27.6 | 30.1 | 30.9 |
| Codex | 30.0 | 37.3 | 39.0 | 29.7 | 37.7 | 39.4 | 24.2 | 24.6 | 25.2 | **36.5** | **42.0** | **42.7** | 33.2 | 37.5 | 39.8 |
| GPT-4 | 32.9 | 40.3 | 43.4 | 24.6 | 26.6 | 27.0 | 34.8 | 41.3 | 45.0 | **41.1** | **48.1** | **49.3** | 34.9 | 41.7 | 45.6 |

statements, we manually check each formalization result. For the miniF2F dataset, the correctness is automatically derived by checking the symbolic equivalence between the formalization result and the provided oracle using ATPs. In the experiments, the number of generations ($k$) is fixed at 10, as we observe that improvements in pass@$k$ (see Figure 2) become marginal with more generations.

**Baseline.** In our experiments, we compare our methods, symbolic equivalence (SymEq), semantic consistency (SemCo), as well as combination strategy (log-comb, linear-comb, and quad-comb), with one baseline and two alternatives. The baseline method uses the log-probability predicted by LLMs to score the $k$ autoformalization candidates. For Codex and GPT-4, which do not provide access to log-probability, they are prompted to rank the candidates instead. We also introduce two additional methods as the alternatives, i.e., a naïve strategy that filters the candidates by whether the ATPs can prove the formalization, and a clustering method that applies the adaptive $k$-means algorithm [72] on BERT embeddings of formal statements.

**Implementation.** For the SymEq method, we implement an equivalence checker as well as peripheral logic based on scala-isabelle [73]. Specifically, the equivalence checker integrates 12 tactics (i.e., `auto`, `simp`, `eval`, `smt`, `blast`, `fastforce`, `force`, `arith`, `linarith`, `presburger`, (`auto simp:field_simps`), and `sledgehammer[timeout=300s]`) provided in Isabelle/HOL [74], as well as two SMT solvers Z3 [29] and CVC5 [30]. For the SemCo method, we use the pretrained BERT [68] to compute the embedding of the informal statement.

## 4.2 Empirical Results

**RQ1: Efficacy.** We compute $n@k$ results for $n = 1, 2, 3$ and five LLMs on the two datasets. As shown in Table 1, SymEq demonstrates superior performance on all cases. For Codex and GPT-4, SymEq achieves the best $n@k$ accuracy with 46.1% (Codex) at $n = 1$ on the MATH dataset, and 41.1% (GPT-4) at $n = 1$ on the miniF2F dataset. For the two smaller LLMs Mistral-7B and Llemma-34B, SymEq also exhibits a notable improvement in 1@$k$ accuracy, surpassing the competitors by at least 4.9%. As for the recently released LLM DeepSeek-v2, SymEq is still effective, resulting in 1@$k$ improvements of 5.1% and 0.8% on the two datasets, respectively.

For SemCo, it successfully achieves the best performance in three cases (3@$k$ of Mistral-7B, 2@$k$ of Llemma-34B, and 3@$k$ of GPT-4) on the MATH dataset. Compared to the baseline, SemCo also performs an improvement, ranging from 0.8% to 3.9% in 1@$k$ accuracy. Compared with the alternatives, SemCo is still slightly more effective, wining 4 out of 5 cases (except for GPT-4) in 1@$k$ accuracy. On the miniF2F dataset, although the improvement of SemCo is narrower, it still wins the alternatives 4 out of 5 cases, and achieves equal results for the rest (DeepSeek-v2) in 1@$k$ accuracy. However, SemCo is much less effective than SymEq in most cases, as it does not grasp the logical nature of formal statements.

**RQ2: Synergy.** We first conduct a detailed analysis of the performance of SymEq and SemCo across different categories of the MATH dataset. The results presented in Table 2 reveal an interesting finding: SymEq and SemCo demonstrate distinctly different performances for each category. This

Table 2: Performance ($1@k$) of our methods (SymEq and SemCo) across various categories from MATH dataset. The formalization results are generated by GPT-4, and the best performance is in bold. The results show that SymEq and SemCo exhibit different behaviors on various categories.

| CATEGORY | #PROBS | SYMEQ | SEMCO |
|---|---|---|---|
| Algebra | 102 | 57.8 | **59.8** |
| Counting and Probability | 46 | **36.9** | 30.3 |
| Geometry | 32 | **28.1** | 25.1 |
| Intermediate Algebra | 77 | **31.1** | 25.9 |
| Number Theory | 42 | 33.3 | **38.0** |
| Prealgebra | 62 | **51.6** | 40.3 |
| Precalculus | 39 | 33.3 | **35.8** |

Figure 4: Performance curve of log-comb for different values of $\alpha$. The formalization results are generated by GPT-4. The results show that the combination can further improve the auto-formalization accuracy with a large sweet spot.

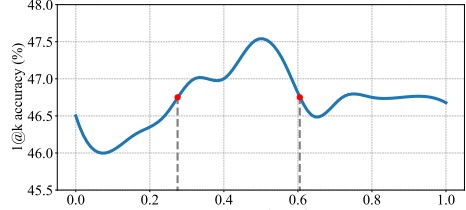

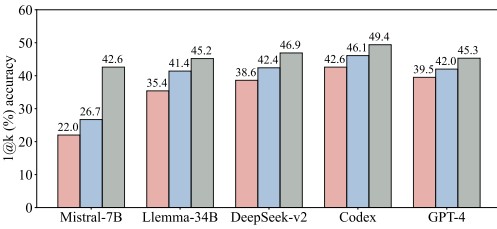 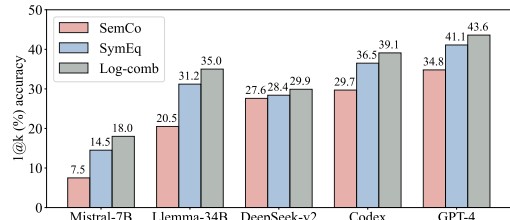

Figure 5: The performance of our proposed combination strategy (log-comb) on the MATH (left) and miniF2F (right) datasets. The results show that the log-comb further boost the autoformalization performance across various LLMs on the two datasets.

variation likely stems from the differing nature and requirements of autoformalizing problems in different categories. For example, geometry problems are more sensitive to semantic consistency since they often involve the translation of visual images, while number theory problems pose greater challenges for checking symbolic equivalence.

Therefore, combining SymEq and SemCo (i.e., log-, linear-, and quad-comb) to further improve autoformalization accuracy is reasonable. We explore the optimal setting for the hyperparameter $\alpha$. In particular, we compute the $1@k$ results of log-comb for various values of $\alpha$ on the MATH dataset, and plot the performance curve in Figure 4. The results demonstrate that the combination strategy can further improve autoformalization accuracy, with a large sweet spot for $\alpha$ ($0.32 - 0.6$).

The performance curves for the other two combination strategies (linear-comb and quad-comb) are provided in Appendix D. We observe that log-comb is more effective and stable than the other two strategies. Furthermore, we fix $\alpha = 0.5$ based on the performance curve and present the overall performance ($n@k$) of log-comb in Figure 5. The results show that log-comb consistently improves autoformalization accuracy across various LLMs on the two datasets, by ranging from 2.3% to 22.6%. Particularly, compared to SymEq, even for the most powerful model GPT-4, log-comb can still further boost the $1@k$ accuracy by 3.3% on the MATH dataset and by 2.5% on the miniF2F dataset. For the Mistral-7B, log-comb presents significant improvements, i.e., 22.6% and 10.5%, respectively.

**RQ3: Labeling-efficiency.** We define average labeling cost for given $k$ autoformalization candidates: $\sigma = \left( \sum_{n=2}^{k-1} n \cdot (n@k - (n-1)@k) \right) + (1 - k@k)$. Based on the average labeling cost $\sigma$, for $N$ mathematical problems, the total number of formal statements to be labeled can be computed by $\sigma N$. Subsequently, we introduce the relative efficiency of two methods as $E = 1 - \sigma/\tilde{\sigma}$. By using the baseline as a reference ($\tilde{\sigma}$), we compute the relative efficiency of each method in Table 3.

It can be observed that our methods, especially log-comb, achieve higher labeling-efficiency compared to the alternatives. On the MATH dataset, log-comb achieves the relative efficiency ranging from 17.7% to 21.6%, with up to three times improvement (on Mistral-7B) than the Cluster method. On the miniF2F dataset, log-comb is still very efficient, by using GPT-4, the relative efficiency achieves 21.9%, outperforming Cluster by 9.2%, SymEq by 3.6%, and SemCo by 5.4%, respectively.

Table 3: Relative efficiency (%) of our methods (SymEq, SemCo, and Log-comb) and alternatives (Naïve, and Cluster) on MATH and miniF2F datasets. The best performance is in bold. Note that the negative results achieved by Naïve are reasonable since it is less effective compared to the baseline. The results show that our proposed methods exhibit higher efficiency enhancement.

| Methods | Naïve | | Cluster | | SymEq | | SemCo | | Log-comb | |
|---|---|---|---|---|---|---|---|---|---|---|
| Dataset | MATH | miniF2F | MATH | miniF2F | MATH | miniF2F | MATH | miniF2F | MATH | miniF2F |
| Mistral-7B | -14.2 | 1.5 | 5.6 | 2.6 | 12.9 | 6.9 | 12.6 | 4.6 | **21.6** | **8.4** |
| Llemma-34B | -14.2 | -4.1 | 10.4 | 4.6 | 15.2 | 15.8 | 14.3 | 8.4 | **18.9** | **19.5** |
| DeepSeek-v2 | -30.9 | -6.9 | 17.2 | 8.3 | 18.7 | 8.0 | 16.3 | 6.8 | **20.5** | **10.0** |
| Codex | -31.4 | -5.1 | 13.7 | 10.4 | 15.3 | 9.6 | 13.5 | 13.6 | **19.9** | **15.3** |
| GPT-4 | -16.9 | -7.0 | 15.6 | 12.7 | 16.3 | 18.3 | 14.7 | 16.5 | **17.7** | **21.9** |

Table 4: Performance ($1@k$) across various difficulty levels from the MATH dataset, with formalization results generated by GPT-4. The results show that autoformalization accuracy is significantly influenced by the difficulty of the problem.

Figure 6: Performance of SymEq using different ATP settings, with formalization results generated by GPT-4. The results indicate that the performance improvement is very narrow by increasing the capability of ATPs.

| Diff.[†] | Baseline | SymEq | SemCo | Log-comb |
|---|---|---|---|---|
| 1 (37) | 64.8 | 67.5 | 64.8 | 75.6 |
| 2 (70) | 44.2 | 47.1 | 47.1 | 52.8 |
| 3 (91) | 48.3 | 57.1 | 47.2 | 53.8 |
| 4 (91) | 26.3 | 34.0 | 31.8 | 40.6 |
| 5 (111) | 24.3 | 24.3 | 26.1 | 27.0 |

[†] $a(b)$ refers to the difficulty level (# problems).

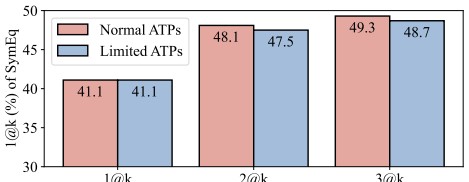

**RQ4: Scalability.** As illustrated by our experimental results in Table 1 and Figure 5, a more powerful LLM, such as GPT-4, often exhibits better autoformalization performance. We provide an additional evidence by examining the performance differences across various difficulty levels in the MATH dataset. The results shown in Table 4 reveal a significant gap in autoformalization performance between Levels 1-3 and Levels 4-5. Hence, the difficulty of the problem is highly correlated with the autoformalizaiton performance, suggesting that an LLM with stronger mathematical reasoning capabilities is more effective in this task.

To investigate the impact of ATP capability, we conduct an additional ablation study. Specifically, we build a limited equivalence checker, in which only two tactics in Isabelle/HOL (`auto` and `simp`) are reserved and the other tactics and SMT solvers are removed. The results are shown in Figure 6, which illustrate that the performance improvement of SymEq is minimal when using ATPs with stronger capability. One possible reason is that, although normal ATPs can prove more symbolic equivalences (2.13 vs. 2.33 per problem on average) than the limited version, this is still not enough to have a major impact on the final symbolic equivalence score.

## 5 Conclusion

In this paper, we present a new framework for improving the autoformalization performance of LLMs. Our techniques address the inherent challenges in autoformalization by overcoming the limitations of traditional self-consistency methods, which struggle to cope with the variance in LLM outputs. Specifically, our framework achieves this goal by combining symbolic equivalence, which grasps the logical nature among formal statements, with semantic consistency, which inspects the semantic coherence between the re-informalization result and the original text. Empirical evaluation on the MATH and miniF2F datasets demonstrates a new level of autoformalization accuracy. Furthermore, our quantitative and case analysis elaborates on the limitations of current LLMs and automatic theorem provers in the task of autoformalization, shedding light on directions for future optimization.

The future directions for our proposed framework could include: (1) Method Adaptation: Extending the framework to support additional theorem provers, such as Lean 4; (2) Model Enhancement: Integrating more advanced or specifically fine-tuned LLMs like ProofGPT [20] and MMA [21] to further enhance the framework's performance; (3) Data Synthesis: Generating higher-quality, aligned informal and formal datasets using the framework. A detailed discussion of the limitations and broader impacts can be found in Appendix A and Appendix B.

## Acknowledgment

We appreciate the anonymous reviewers for their valuable insights and helpful comments. This work is supported by the National Natural Science Foundation of China (Grants #62025202) and the Frontier Technologies R&D Program of Jiangsu (BF2024059). Xian Zhang (`zhxian@microsoft.com`) and Xiaoxing Ma (`xxm@nju.edu.cn`) are the corresponding authors.

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

## A  Boarder Impact

The paper focuses on the formalization of mathematical theorems through large langauge models. There are many potential societal consequences of our work, and we firmly believe that the majority of these impacts are positive and none which we feel must be specifically highlighted here.

## B  Limitations of this work

Though achieving a considerable improvement in the autoformalization task, we also find some remaining challenges either rooted in current LLMs, ATPs, semantic embedding or evaluation metrics:

**LLMs lack the knowledge of formal library.** As shown in Figure 2, even with a sufficiently large $k$, the LLMs may still not generate one correct formalization for some problems due to not knowing the existing functions or definitions in formal library for the concepts, which necessitate the need for advanced retrieval mechanism or more translation pairs with finetuning.

**Autoformalization is more than translation.** Except for the exact mapping from natural language concepts to existing formal language functions, some math problems require the combination or the variants of certain standard definitions or functions in the formal library, which further necessitate LLMs to be capable of some basic reasoning or modeling. This again exceeds the capability of current LLMs, as indicated by Table 4.

**ATPs are not strong enough for automation.** According to McCarthy's classical idea [1], an automatic process to evaluate the correctness of formalization can be to prove the formalization with ATPs. If proved, it is with high possibility the formalization is correct. However, As shown in Table 1, current ATPs are far from the capability to prove the math problems of the high school level. Furthermore, ATPs are even incapable to generate the proof of symbolic equivalence for some problems, which is often much easier than the proof of the original problems.

**Embeddings may neglect the nuances in natural language.** When using embeddings to check semantic consistency, we assume embeddings can reflect the differences between the informalized statement and the original. However, there are many nuances in math statements that even a single notation change can result in totally different semantics. Current embeddings may not differentiate the minor but significant change.

**Evaluation still requires the grounding effort of humans.** Even with an ideal ATP and an ideal embedding mapping, the evaluation of the formalization still requires the finalization of humans. As shown in Example 13, from any perspective (either symbolically or semantically), the formalization can be marked as correct. But from the perspective of humans, the formalization oversimplifies the problem or models the problem based on an abstraction level uncommon for humans. Therefore, human preferences are still essential for evaluating the formalizations.

## C  Two examples of variable matching

In Example 1, we show how to standardize the formal statement whose proof goal is represented as the numerical equality. We introduce a new variable $\alpha$, transforming two formal statements into $(x = 2/3 \land y = 6 \land \alpha = x * y - 4) \to (\alpha = 0)$ and $(a = 2/3 \land b = 6 \land \alpha = a * b - 4) \to (\alpha = 0)$. Next, by viewing $x, y, a, b$ as auxiliary variables, the symbolic equivalence $(x = 2/3 \land y = 6 \land \alpha = x * y - 4) \equiv (a = 2/3 \land b = 6 \land \alpha = a * b - 4)$ and $(\alpha = 0) \equiv (\alpha = 0)$ can be easily proved.

In Example 2, since the proof goal is not numerical inequality, we have to build the bipartite graph with nodes $(n, a, b, c, d, e, x, y, z, u, v)$. The edge weights are computed by string edit distance and we can derive the final variable mapping $a \leftrightarrow x, b \leftrightarrow y, c \leftrightarrow z, d \leftrightarrow u, e \leftrightarrow v$, using the maximum bipartite matching. Finally, we finish the standardization by refreshing the variable notations.

## D  Performance curves of various combination strategies

We provide performance curves of linear-comb and quad-comb in Figure 7. Compared with log-comb, we observe that linear-comb and quad-comb have smaller effective range of $\alpha$. Therefore, we select the log-comb strategy and fix $\alpha = 0.5$ in subsequent experiments.

> **Example 1: An Illustrative Example of Variable Matching**
>
> **Prompt:**
> Natural language version: "Find the product of $0.\backslash\backslash overline6$ and 6. The answer is 4".
> Translate the natural language version to an Isabelle version.
>
> ---
>
> **GPT-4 output (No.2):**
>
> ```
> theorem
> fixes x y :: real
> assumes "x = 2/3"
> and "y = 6"
> shows "x * y = 4"
> ```
>
> **Standardized GPT-4 output (No.2):**
>
> ```
> theorem
> fixes x y alpha :: real
> assumes "x = 2/3"
> and "y = 6"
> and "alpha = x * y - 4"
> shows "alpha = 0"
> ```
>
> **GPT-4 output (No.3):**
>
> ```
> theorem
> fixes a b :: real
> assumes "a = 2/3"
> and "b = 6"
> shows "a * b = 4"
> ```
>
> **Standardized GPT-4 output (No.3):**
>
> ```
> theorem
> fixes a b alpha :: real
> assumes "a = 2/3"
> and "b = 6"
> and "alpha = a * b - 4"
> shows "alpha = 0"
> ```

> **Example 2: An Illustrative Example of Variable Matching**
>
> **Prompt:**
> Natural language version: "Prove that 4 raised to the power of the sum of five positive integers is divisible by 2". Translate the natural language version to an Isabelle version.
>
> ---
>
> **GPT-4 output (No.1):**
>
> ```
> theorem
> fixes n a b c d e :: nat
> assumes "n =
>         a + b + c + d + e"
> shows  "2 dvd (4^(n))"
> ```
>
> **Standardized GPT-4 output (No.1):**
>
> ```
> theorem
> fixes n v1 v2 v3 v4 v5 :: nat
> assumes "n =
>         v1 + v2 + v3 + v4 + v5"
> shows  "2 dvd (4^(n))"
> ```
>
> **GPT-4 output (No.2):**
>
> ```
> theorem
> fixes x y z u v :: nat
> shows  "2 dvd
> (4 ^ (x + y + z + u + v))"
> ```
>
> **Standardized GPT-4 output (No.2):**
>
> ```
> theorem
> fixes v1 v2 v3 v4 v5 :: nat
> shows  "2 dvd
> (4 ^ (v1 + v2 + v3 + v4 + v5))"
> ```

# E   Some motivation examples

We demonstrate some typical errors when using GPT-4 to formalize the mathematical statement. In particular, Example 3 illustrates that GPT-4's hallucination leads to the generation of a non-existent concept (`reflection_matrix`). In Example 4, GPT-4 formalizes the condition "$r$, $s$, and $t$ be the roots of the equation $4x^3 - 59x^2 + 32x - 32 = 0$. " by using Vieta's formulas, but the formulation is incorrect ($r * s * t = 8$) and a condition ($r * s + r * t + t * s = 8$) is ignored. As to Example 5, GPT-4 should determine whether to use `powr` or `^`. In Isabelle language, `^` is only applicable to natural number exponents, but it is a real number in the example.

We also highlight some cases (i.e., Example 6 to Example 13) when reviewing the autoformalization results, e.g., incorrect formalization oracle, incorrect label, strange failure in checking symbolic equivalence, and so on, which shed light on the following messages:

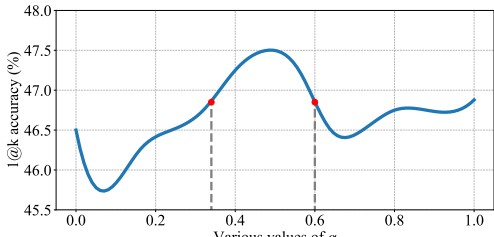 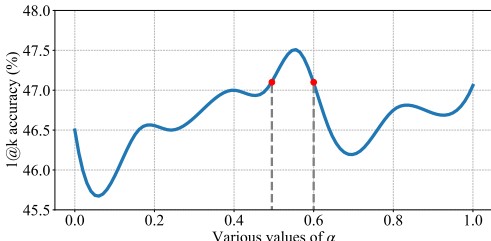

Figure 7: Performance curves of linear- (left) and quad- (right) comb across various values of $\alpha$. The formalization results are generated by GPT-4. The results show that both combination strategies successfully improve autoformalization accuracy, while the effective range of quad-comb is smaller.

**The difficulty of human labeling**. We find people even experts are prone to flaws in writing or labeling formalizations. As shown in Example 6, humans may overlook the potential correct candidate while our framework can discover this via symbolic equivalence. As shown in Example 7, the oracle in miniF2F is wrong because when $n$ equals 0, $f\ n = 0$, which makes one assumption of the oracle false. Example 8 also shows another error in miniF2F oracles. In fact, the miniF2F dataset has been revised and checked by groups of experts [19, 10]. Therefore, it is challenging to guarantee the correctness of formalized statements without the scrutinized validations of experts or any assistance from frameworks like ours.

**The difficulty of symbol grounding**. The symbol grounding problem is hard not only for LLMs but also for humans. As shown in Example 9 and Example 10, humans would simply regard $^\wedge$ and $powr$ or $log$ and $ln$ as identical while their underlying symbolic implementations are much different. The most interesting case is Example 11 where spaces around $4$ will make the original Isabelle system behave differently with the type inference, leading to a failure of proving equivalence.

**The (over)simplification with reasoning.** We find that LLMs tend to make one or more steps' reasoning during autoformalization, which would simplify and misalign with the original statement. As shown in Example 12, LLM may implicitly consolidate assumptions. And for concepts not frequently touched in formal language (e.g., geometry), LLM may directly oversimplify the problem as arithmetic, as shown in Example 13.

# F   Prompt examples for autoformalizations

We set up instructions and eight gold examples with both symbolic and semantic correctness to improve the generation performance of GPT-4. Our detailed prompts are shown in Prompt 1 and 2.

---

**Prompt 1: Autoformalization Prompt**

As a mathematician familiar with Isabelle, your task is to translate the natural language problem into an Isabelle language version. \n\n Natural language version:\n{*instruction*}\n\n Translate the natural language version to an Isabelle version:\n

---

**Prompt 2: Informalization Prompt**

As a mathematician familiar with Isabelle, your task is to the Isabelle language problem back to a natural language version. \n\n Isabelle language version:\n{*instruction*}\n\n Translate the Isabelle language problem back to a natural langauge version:\n

---

### Example 3: mathd precalculus 1033

**Prompt:**

Natural language version: "Let $\mathbf{R}$ be the matrix for reflecting over the vector $\begin{pmatrix} 1 \\ 5 \end{pmatrix}$. Find $\det \mathbf{R}$." Translate the natural language version to an Isabelle version.

**GPT-4 output:**

```
theorem
fixes R :: "real^2^2"
assumes h0 : "R = reflection_matrix (vector [1, 5])"
shows  "determinant R = -1"
```

**Cause:** GPT-4 introduces a non-existent concept (`reflection_matrix`).

---

### Example 4: mathd intermediate 745

**Prompt:**

Natural language version: "Let $r$, $s$, and $t$ be the roots of the equation $4x^3 - 59x^2 + 32x - 32 = 0$. Find the value of $f(r) + f(s) + f(t)$, where $f(x) = 4x^3 - 59x^2$". Translate the natural language version to an Isabelle version.

**GPT-4 output:**

```
theorem
fixes x r s t :: real
        and f :: "real \<Rightarrow> real"
assumes h0 : "r * s * t = 32"
and h1 : "r + s + t = 59/4"
and h2 : "\<forall> x. f x = 4 * x powr 3 - 59 * x powr 2"
shows  "f(r) + f(s) + f(t) = -376"
```

**Cause:** GPT-4 ignores a condition ($r*s + r*t + t*s = 8$) when using Vieta's formulas.

---

### Example 5: aime 1984 p5

**Prompt:**

Natural language version: "Determine the value of $ab$ if $log_8 a + log_4 b^2 = 5$ and $log_8 b + log_4 a^2 = 7$. Show that it is 512.". Translate the natural language version to an Isabelle version.

**GPT-4 output:**

```
theorem
fixes a b :: real
assumes h0 : "log 8 a + log 4 (b^2) = 5"
and h1 : "log 8 b + log 4 (a^2) = 7"
shows  "a * b = 512"
```

**Cause:** GPT-4 incorrectly uses ^ to real numbers $a$ and $b$.

### Example 6: mathd algebra 2471

**Prompt:**
Natural language version: "For how many positive integers $x$ is $x^2 + 4x + 4$ between 10 and 50? The final answer is 4.". Translate the natural language version to an Isabelle version.

- - - - - - - - - - - - - - - - - - - - - - - - - - - - - - - - - - - - - - - - - - - - - -

**GPT-4 output (No.1):**

```
theorem
fixes x :: nat and f :: "nat \<Rightarrow> nat"
assumes h0 : "\<forall> x. f x = x^2 + 4*x + 4"
and h1 : "\<forall> x. (10 \<le> f x \<and> f x \<le> 50) = (A x)"
shows "card {x. A x} = 4"
```

**GPT-4 output (No.2):**

```
theorem
fixes x :: nat and f :: "nat \<Rightarrow> nat"
assumes h0 : "\<forall> x. f x = x^2 + 4 * x + 4"
and h1 : "\<forall> x. 10 \<le> f x
\<and> f x \<le> 50 \<longrightarrow> (\<exists> n. x = n)"
shows "card {x. 10 \<le> f x \<and> f x \<le> 50} = 4"
```

> **Cause:** GPT-4 outputs are symbolically equivalent, but human expert only labeled No.1 as correct, and incorrectly overlook No.2.

---

### Example 7: imo 1977 p6

**Prompt:**
Natural language version: "Let $f(n)$ be a function $f : \mathbb{N}^+ \to \mathbb{N}^+$. Prove that if $f(n+1) > f(f(n))$ for each positive integer $n$, then $f(n) = n$". Translate the natural language version to an Isabelle version.

- - - - - - - - - - - - - - - - - - - - - - - - - - - - - - - - - - - - - - - - - - - - - -

**GPT-4 output (No.1):**

```
theorem
fixes f :: "nat \<Rightarrow> nat"
assumes h0 : "\<forall> n > 0. f (n + 1) > f (f n)"
shows  "f n = n"
```

**Oracle:**

```
theorem
fixes f :: "nat \<Rightarrow> nat"
assumes "\<forall> n. f (f n) < f (n + 1)"
and "\<forall> n. f n >0"
shows "\<forall> n. f n = n"
```

> **Cause:** the provided oracle is incorrect

### Example 8: mathd 1485

**Prompt:**
Natural language version: "Find the value of $x$ that satisfies $\frac{1}{3x-1} = \frac{2}{x+1}$". Translate the natural language version to an Isabelle version.

**GPT-4 output (No.1):**

```
theorem
fixes x :: real
assumes h0 : "3 * x - 1 \<noteq> 0"
and h1 : "x + 1 \<noteq> 0"
and h2 : "1 / (3 * x - 1) = 2 / (x + 1)"
shows "x = 3 / 5"
```

**Oracle:**

```
theorem
fixes x :: real
assumes h0 : "(1 / (3 * x - 1)) = (2 / (x + 1))"
shows "x = 3 / 5"
```

**Cause:** implicit condition of division: dividend is not equal to zero

---

### Example 9: aime 1984 p5

**Prompt:**
Natural language version: "Determine the value of $ab$ if $log_8 a + log_4 b^2 = 5$ and $log_8 b + log_4 a^2 = 7$. Show that it is 512.". Translate the natural language version to an Isabelle version.

**GPT-4 output (No.1):**

```
theorem
fixes a b :: real
assumes h0 : "log 8 a + log 4 (b powr 2) = 5"
and h1 : "log 8 b + log 4 (a powr 2) = 7"
shows  "a * b = 512"
```

**GPT-4 output (No.2):**

```
theorem
fixes a b :: real
assumes h0 : "log 8 a + log 4 (b^2) = 5"
and h1 : "log 8 b + log 4 (a^2) = 7"
shows  "a * b = 512"
```

**Cause:** fail to prove equivalence `powr` with `^`

## Example 10: aime 1984 p5

**Prompt:**
Natural language version: "Determine the value of $ab$ if $log_8a + log_4b^2 = 5$ and $log_8b + log_4a^2 = 7$. Show that it is 512.". Translate the natural language version to an Isabelle version.

- - - - - - - - - - - - - - - - - - - - - - - - - - - - - - - - - - - - - - - - - - - - - - - -

**GPT-4 output (No.1):**

```
theorem
fixes a b :: real
assumes h0 : "log 8 a + log 4 (b powr 2) = 5"
and h1 : "log 8 b + log 4 (a powr 2) = 7"
shows  "a * b = 512"
```

**Oracle:**

```
theorem
fixes a b ::real
assumes h0: "(ln a) / (ln 8) + (ln (b^2)) / (ln 4) = 5"
and h1: "(ln b) / (ln 8) + (ln (a^2)) / (ln 4) = 7"
shows  "a * b = 512"
```

**Cause:** fail to prove equivalence `powr` with `^` and $log$ with $ln$

---

## Example 11: numbertheory 2dvd4expn

**Prompt:**
Natural language version: "Show that for any positive integer $n$, 2 divides $4^n$". Translate the natural language version to an Isabelle version.

- - - - - - - - - - - - - - - - - - - - - - - - - - - - - - - - - - - - - - - - - - - - - - - -

**GPT-4 output (No.1):**

```
theorem
fixes n :: nat
shows  "2 dvd (4^n)"
```

**GPT-4 output (No.2):**

```
theorem
fixes n :: nat
shows  "2 dvd (4 ^ n)"
```

**Checking the equivalence of GPT-4 outputs (by *simp):***

```
theorem
fixes var_0 :: nat
shows "{(var_0). (2 dvd ((4::nat)^var_0))}
              = {(var_0). (2 dvd ((4::nat) ^ var_0))}"
```

**Cause:** the symbolic equivalence is trivial for human, but cannot be proved by ATP. The data types of two "4" cannot be inferred to be identical.

## Example 12: mathd algebra 392

**Prompt:**
Natural language version: "The sum of the squares of three consecutive positive even numbers is 12296. Find the product of the three numbers divided by 8. Show that it is 32736.". Translate the natural language version to an Isabelle version.

- - - - - - - - - - - - - - - - - - - - - - - - - - - - - - - - - - - - - - - - - - - -

**GPT-4 output (No.1):**

```
theorem
fixes n :: real
assumes h0 : "n > 0"
and h1 : "n mod 2 = 0"
and h2 : "n powr 2 + (n + 2) powr 2 + (n + 4) powr 2 = 12296"
shows  "(n * (n + 2) * (n + 4)) / 8 = 32736"
```

**GPT-4 output (No.2):**

```
theorem
fixes n :: real
assumes h0 : "n > 0"
and h1 : "(2 * n) powr 2 + (2 * n + 2) powr 2
                       + (2 * n + 4) powr 2 = 12296"
shows  "((2 * n) * (2 * n + 2) * (2 * n + 4)) / 8 = 32736"
```

**Cause:** fail to prove the equivalence with $n \bmod 2$ and $2 * n$

---

## Example 13: mathd geometry 710

**Prompt:**
Natural language version: "For triangle $ABC$, points $D$ and $E$ are the midpoints of sides $AB$ and $AC$, respectively. Side $BC$ measures six inches. What is the measure of segment $DE$ in inches?
[asy] draw((-5,-5)–(10,3)–(0,10)–cycle); draw((-2.5,2.5)–(2.5,-1)); label("$A$",(-5,-5),SW); label("$B$",(0,10),N); label("$C$",(10,3),SE); label("$D$",(-2.5,2.5),NW); label("$E$",(2.5,-1),SE); [/asy] The final answer is 3.". Translate the natural language version to an Isabelle version.

- - - - - - - - - - - - - - - - - - - - - - - - - - - - - - - - - - - - - - - - - - - -

**GPT-4 output (No.1):**

```
theorem
fixes A B C D E :: "real*real"
and dist :: "(real*real) \<Rightarrow> (real*real)
                          \<Rightarrow> real"
assumes h0 : "dist B C = 6"
and h1 : "D = midpoint A B"
and h2 : "E = midpoint A C"
shows "dist D E = 3"
```

**GPT-4 output (No.2):**

```
theorem
fixes DE BC :: real
assumes h0 : "BC = 6"
and h1 : "DE = BC / 2"
shows "DE = 3"
```

**Cause:** fail to formalize geometry problems

