# OpenReview forum: "Autoformalize Mathematical Statements by Symbolic Equivalence and Semantic Consistency"
_NeurIPS.cc/2024/Conference — NeurIPS 2024 poster_

### Official Review · Reviewer_GLbW · 2024-06-13

**Soundness:** 2
**Presentation:** 4
**Contribution:** 2
**Rating:** 5
**Confidence:** 4

**Summary:**

This paper designs two self-consistency methods for formalizing mathematical statements in natural language into Isabelle --- symbolic equivalence and semantic consistency. In particular, symbolic equivalence measures whether the formalized statements are equivalent (judged by the Isabelle’s built-in automated theorem provers), and semantic consistency measures the similarity of the natural language translation of the formalized statement and its original statement (In other words, semantic consistency measures whether the formalized statement keeps the semantic meaning.) Tested with various base models, this paper shows that the combination of the two proposed methods improves the success rate of the autoformalization process.

**Strengths:**

-	This paper studies an interesting topic, self-consistency for autoformalization, which is non-trivial because the target is not unique. To the best of my knowledge, this is the first paper to study the self-consistency methods for autoformalization.
-	Figure 2 shows that self-consistency methods for autoformalization have great potential because the pass@k accuracy keeps increasing with more trails.

**Weaknesses:**

-	The major weakness of this paper is the scope of the autoformalization --- the symbolic equivalence method only applies to the formalization of theorem statements instead of their proofs. Autoformalization of theorem statements is an arguably less significant problem since it cannot be used to verify any informal proofs.

**Questions:**

-	Definition 1 looks unnecessarily complicated to me. For two theorems A and B, does the symbolic equivalence just mean that A and B are equivalent (which can be checked by the Isabelle solve_direct tool)?
-	The variable misalignment example is confusing to me. In particular, do the authors suggest that the No. 2 and No.3 outputs in Figure 1 should be equivalent? It seems to me that in output No. 3, the variables x and y are untyped but x, y in output No. 2 have type real. In addition, it is even unclear to me whether output No. 3 is provable.
-	Is the phenomenon in Figure 2 true for models finetuned for autoformalization such as MMA? In addition, does SymEq/SemCo still improve the performance over MMA models?

**Limitations:**

The authors adequately addressed the limitations and potential negative societal impact.

---

> ### Author Rebuttal · Authors · 2024-08-07
>
> **Dear Reviewer GLbW**
>
> Thank you for your review and detailed feedback. In the following, please let us address the concerns and questions you've pointed out.
>
> **[Weakness #1] The scope of our framework and the significance of statement autoformalization**
>
> First, we would like to clarify that, the symbolic equivalence could be adapted to the proof autoformalization. Formally speaking, a formal proof is nothing but a sequence of formal subgoals logically connected (e.g., see Figure 2 in DSP [1]). Since there is no clear difference between the autoformalization for the final goal and the intermediate subgoals, the symbolic equivalence should be still effective in proof autoformalization. We leave the exploration of this direction as the future work.
>
> Next, while we do recognize the importance of automatic verification of informal proofs and regard it as our future work, we respectfully disagree with the comment that “Autoformalization of theorem statements is an arguably less significant problem since it cannot be used to verify any informal proofs”. Elaborately, automatic verification of informal proofs can be achieved by autoformalizing each subgoals in the informal proof, and the validity of the formal proof can be verified with a proof assistant such as Isabelle or Lean. Hence, we do believe the statement autoformalization capability is the foundation for verifying informal proofs.
>
> Moreover, technically, proof autoformalization requires not only the informal statement and proof as input but also the formal statement [1, 2]. Therefore, having a correct formal statement for the informal statement is a crucial precondition for the success of proof autoformalization.
>
> Finally, for theorem proving, proof autoformalization is not necessary. For example, the recent AlphaProof project [3] by Google DeepMind uses statement autoformalization to generate approximately 100 million problems and trains its solver network using the AlphaZero algorithm.
>
> [1] Draft, Sketch, and Prove: Guiding Formal Theorem Provers with Informal Proofs. ICLR 2023
>
> [2] Don't Trust: Verify -- Grounding LLM Quantitative Reasoning with Autoformalization. ICLR 2024
>
> [3] https://deepmind.google/discover/blog/ai-solves-imo-problems-at-silver-medal-level/
>
> **[Question #1] Compared with solve_direct tactic**
>
> It is quite hard to formally define the equivalence between two theorems, and hence we try to define the symbolic equivalence as a metric of the consistency between two formal statements. Continue the example shown in Figure 1, given a formal statement,`solve_direct` can successfully detect its duplicate version, but cannot detect its variant although it is still a correct formalization.
>
> ```bash
> theorem first_statement :
> fixes a b :: real
> assumes "a = 2/3"
> and "b = 6"
> shows "a * b = 4"
>   using assms(1) assms(2) by force
>
> theorem second_statement :
> fixes x y :: real
> assumes "x = 2/3"
> and "y = 6"
> shows "x * y = 4"
>   solve_direct
>
> (* solve_direct: the current goal can be solved directly with
>   Scratch.first_statement: ?a = 2 / 3 ⟹ ?b = 6 ⟹ ?a * ?b = 4 *)
>
> theorem first_statement :
> fixes a b :: real
> assumes "a = 2/3"
> and "b = 6"
> shows "a * b = 4"
>   using assms(1) assms(2) by force
>
> theorem second_statement :
> fixes y :: real
> assumes  "y = 6"
> shows "(2 / 3) * y = 4"
>   solve_direct
>
> (* No proof found *)
> ```
>
> Empirically, we also replace the symbolic equivalence with `solve_direct` tactic and conduct a comparison experiment using GPT-4 autoformalization results. The 1@k results shown in the following indicate that the symbolic equivalence still outperforms `solve_direct` either when solely used or combined with the semantic consistency.
>
> | Dataset | solve_direct | SymEq | Log-comb (solve_direct + SemCo) | Log-comb (SymEq + SemCo) |
> | --- | --- | --- | --- | --- |
> | MATH | 37.5 | 42.0 | 42.2 | 45.3 |
> | miniF2F | 34.6 | 41.1 | 39.1 | 43.6 |
>
> **[Question #2] A mistake in Figure 1**
>
> We apologize for the mistake made in Figure 1. The No.3 output in Figure 1 is provable and it should be
>
> ```bash
> theorem
> fixes x y :: real
> assumes "x = 2/3"
> and "y = 6"
> shows "x * y = 4"
> ```
>
> **[Question #3]  Is the proposed methods still effective on MMA fine-tuned models?**
>
> Thanks for the comment. We trained a Mistral 7B model using the Isabelle data in MMA, and evaluate our framework using this model on 60 MATH problems. The pass@k (k=1,2,3,10) are 23.3%, 31.6%, 35%, 46.6%, respectively, indicating the curve in Figure 1 is still effective for this model. In addition, the n@k results are provided as follows, and we can observe that our framework also works well.
>
> | 1@k |  |  |  |
> | --- | --- | --- | --- |
> | Baseline | SymEq | SemCo | Log-comb |
> | 23.3 | 26.6 | 23.3 | 26.6 |
>
> | 2@k |  |  |  |
> | --- | --- | --- | --- |
> | Baseline | SymEq | SemCo | Log-comb |
> | 31.6 | 35.0 | 33.3 | 36.6 |
>
> | 3@k |  |  |  |
> | --- | --- | --- | --- |
> | Baseline | SymEq | SemCo | Log-comb |
> | 35.0 | 36.6 | 35.0 | 38.3 |

---

> > ### Comment · Reviewer_GLbW · 2024-08-11
> >
> > I thank the authors for the examples and additional results. My concerns are addressed and I will increase my score accordingly.

---

### Official Review · Reviewer_LZ2a · 2024-06-29

**Soundness:** 3
**Presentation:** 3
**Contribution:** 3
**Rating:** 6
**Confidence:** 4

**Summary:**

This paper targets the task of autoformalization for mathematics, which inputs an informal theorem statement (in English) and outputs a formal theorem statement (inside an interactive theorem prover). Prior work has used LLMs in-context to produce autoformalizations. The work is motivated by a disparity between top-1 accuracy and top-k accuracy when sampling formalizations from LLMs. The proposed method works atop any generative model of formalized statements. Their approach is to refine the k samples to a final choice via a voting mechanism. The approach is twofold - first a symbolic equivalence method uses automated theorem provers (ATPs) to check for equivalence between two formalized statements. The k-samples are then clustered by equivalence, and then they perform semantic consistency: measuring the embedding similarity between the original informal statement and the informalized formal statement. They indicate improvements over the baselines.

**Strengths:**

1. The idea to compare formalization candidates via symbolic equivalence is interesting. It is a nice strategy to impose additional structure on the k-autoformalization samples. The use of the semantic consistency is a natural extension of the distilled backpropogation idea presented in (https://arxiv.org/abs/2302.12433). While measuring alignment with the informal statement is not automatable, the symbolic equivalence is a nice intermediate which I believe will be further explored in the future.

2. The results indicate that their approach is effective. As the approach can be placed atop any autoformalization strategy sampling > 1 candidates from a generative model, the idea is generally applicable and should be useful for all such future work.

**Weaknesses:**

1. The semantic consistency alone does not appear to perform much better, I wonder if this is due to the choice of BERT for the embeddings. Perhaps you can use a newer model to judge its efficacy?

2. The formulae for combining the semantic consistency score do not seem to have any motivation for the particular choice of $f(x) = x, x^2, \log x$. Can the authors explain the rationale for these choices?

3. The definition of $\tau$-semantic consistency is not used elsewhere. In particular, it is unclear the purpose of the incorporation of $\tau$ as it is not mentioned again.

**Questions:**

1. The BERT model is used for measuring embedding similarity. How do the results change if using better models? BERT is quite outdated at this point.

2. Can you provide analysis as to why the log-combination is better than the other two?

3. On line 308 you indicate that on average about 2 symbolic equivalences are proven for k = 10 samples, this is lower than I had expected. What is the average cluster size per problem? Is it often the case that the models output candidates which are not symbolically equivalent (or at least, cannot be proven to be)? Does the chosen formalization end up coming from a larger cluster, even with semantic consistency being used?

4. In algorithm 1 you mentioned you standardize candidates $\Psi_i, \Psi_j$. What is the protocol for performing this operation? Do all hypotheses go in P_i, P_j?

Typos:
1. On line 164 you mentioned that prior works commonly assign scores based on cluster size. Could you include a reference?
2. On line 166 you mentioned the self-consistency is the similarity between the autoformalized version and its informalization. Did you mean the informalized version and the informal statement?
Line 34: Cover -> Recover?
Line 261: wining -> winning
Line 301: autoformalizaiton -> autoformalization
Line 578: "Some motivation examples -> "Some motivational examples" or "Motivating examples"

**Limitations:**

I believe the authors have adequately addressed the potential limitations of their work in the limitations section in the appendix.

---

> ### Author Rebuttal · Authors · 2024-08-07
>
> **Dear Reviewer LZ2a**
>
> Thank you for the insightful feedback on our paper.  We appreciate the time and effort you have put into reviewing our work, and we are grateful for encouraging comments such as interesting idea, effective results, and general applicability. We have carefully read your review and addressed your concerns as follows.
>
> **[Weakness #1 & Question #1] Newer model for the semantic consistency**
>
> Thanks for the comment. As suggested, we use e5-Mistral [1] instead of the BERT as a new embedding model. The results are shown as follows. It appears that the embedding model update makes a minimal impact on performance. This may due to that the effectiveness of informalization may be more critical than the embedding model for the semantic consistency.
>
> | MATH |  |  |  | miniF2F |  |  |  |
> | --- | --- | --- | --- | --- | --- | --- | --- |
> | SemCo(BERT) | SemCo(e5-Mistral) | Log-comb (BERT) | Log-comb (e5-Mistral) | SemCo(BERT) | SemCo(e5-Mistral) | Log-comb (BERT) | Log-comb (e5-Mistral) |
> | 39.5 | 40.5 | 45.3 | 46.7 | 34.9 | 34.4 | 43.6 | 42.6 |
>
> **[Weakness #1 & Question #2] The rationale of combination strategy**
>
> The choice of combination strategy (linear, quadratic, or logarithmic) is determined empirically. Curves of three combination strategies are shown in Figure 4 and Figure 7, and we can observe that the log-combination is the most stable. A potential reason for this result may due to that the log transformation numerically imposes a heavier penalty on candidates with lower scores, thereby reducing their likelihood of being selected. We will present a comprehensive analysis about combination strategy in our revision.
>
> **[Weakness #3] The definition of $\tau$-semantic consistency**
>
> The definition of $\tau$-semantic consistency is included for the completeness and serves as a counterpart to the symbolic equivalence. In the implementation, we directly use the embedding similarity as the score of semantic consistency.
>
> **[Question #3.1] What is the average cluster size per problem?**
>
> We apologize for the confusion. In the paper (line 308), “2.13 vs. 2.33” is computed by the average size of symbolic equivalence classes across ALL problems, i.e., $\sum$ sizes of all classes on all problems / $\sum$ Number of classes on all problems. For example, given two problems and their symbolic class sizes are 10 and 1 (i.e., all problem are symbolically equivalent and none problem are symbolically equivalent). Then the average size is calculated by (10 + 10 $\times$ 1) / (1 + 10). Following your suggestion, we list the average and maximum cluster size per problem (using GPT-4 outputs) as follows. We will clarify this and include the results in the revision.
>
> | Dataset | MATH | miniF2F |
> | --- | --- | --- |
> | Average cluster size per problem | 3.19 | 4.41 |
> | Max cluster size per problem | 4.89 | 6.11 |
>
> **[Question #3.2] Is it often the case that the models output candidates which are not symbolically equivalent?**
>
> From the above results, the model frequently produces candidates that are symbolically equivalent.
>
> **[Question #3.3] Does the chosen formalization end up coming from a larger cluster, even with semantic consistency being used?**
>
> Yes. Generally, the efficacy of log-combination is derived from that (1) the symbolic equivalence determines multiple large clusters, and then (2) the semantic consistency distinguishes the best one from them.
>
> **[Question #4] The protocol for performing standardization**
>
> The standardization is employed to align variables and it does put all hypothesis into $P_i$ and $P_j$. Elaborately, given two formal statements, if their conclusions are in the form of the numerical relation, the standardization introduces new variables $\alpha$, and identifies other variable as auxiliary variables. For other cases, the standardization conduct the bipartite matching for the alignment.
>
> **[Typos #1]**
>
> Thanks for the comments. We will include Self-Consistency [2], CodeT [3], and Multi-Perspective Self-Consistency [4], as references.
>
> **[Typos #2]**
>
> Yes, the self-consistency in the context refers to the similarity between the informalized version and the informal statement.
>
>  [1] https://huggingface.co/intfloat/e5-mistral-7b-instruct
>
>  [2] Self-Consistency Improves Chain of Thought Reasoning in Language Models, ICLR 2023
>
>  [3] CodeT: Code Generation with Generated Tests, ICLR 2023
>
>  [4] Enhancing Large Language Models in Coding Through Multi-Perspective Self-Consistency, ACL 2024

---

> > ### Comment · Reviewer_LZ2a · 2024-08-10
> >
> > Thanks to the authors for addressing my questions.
> >
> > >The standardization is employed to align variables and it does put all hypothesis into $P_i$ and $P_j$. Elaborately, given two formal statements, if their conclusions are in the form of the numerical relation, the standardization introduces new variables $\alpha$, and identifies other variable as auxiliary variables. For other cases, the standardization conduct the bipartite matching for the alignment.
> >
> > Is it normally the case that the symbolic equivalence is only checking for statements being equivalent up to variable naming, but no more than that? What is an instance where the symbolic equivalence checks for a more nontrivial property? I believe the ones included in the main body and the appendix are all instances of variable matching.

---

> ### Author Response · Authors · 2024-08-10
> **Response to the Reviewer LZ2a**
>
> Equivalence checking solely up to variable naming is not considered the general case. In fact, the primary challenge in autoformalization lies in accurately modeling mathematical concepts. Therefore, symbolic reasoning that extends beyond mere variable matching is essential when language models represent the same concept in various ways.
>
> We list four examples in the following.
> ```
> (*A standard six-sided die was rolled 50 times, and the outcomes are shown in the table. What is the average of the 50 outcomes? Express your answer as a decimal to the nearest hundredth. \\begin{tabular}{|c|c|}\n\\hline\nOutcome&$\\#$ of Occurrences\\\\\\hline\n1&14\\\\\\hline\n2&5\\\\\\hline\n3&9\\\\\\hline\n4&7\\\\\\hline\n5&7\\\\\\hline\n6&8\\\\\\hline\n\\end{tabular}*)
>
> theorem
> fixes outcome :: "nat \<Rightarrow> nat"
> assumes h0 : "outcome 1 = 14"
> and h1 : "outcome 2 = 5"
> and h2 : "outcome 3 = 9"
> and h3 : "outcome 4 = 7"
> and h4 : "outcome 5 = 7"
> and h5 : "outcome 6 = 8"
> and h6 : "sum outcome {1..6} = 50"
> shows "(sum (\<lambda>i. i * outcome i) {1..6}) / 50 = 3.24"
>
> theorem
> fixes avg :: real
> assumes h0 : "avg = (14*1 + 5*2 + 9*3 + 7*4 + 7*5 + 8*6) / 50"
> shows "avg = 3.24"
>
> (* Problem : The sum of the $x$-coordinates of the vertices of a triangle in the Cartesian plane equals $\\sqrt{13}$.
> 	Let $S$ equal the sum of the $x$-coordinates of the midpoints of the sides of the triangle.
> 	Find $S^2$.*)
>
> theorem
> fixes A B C M1 M2 M3 :: "real * real"
> assumes h0 : "fst A + fst B + fst C = sqrt 13"
> and h1 : "M1 = ((fst A + fst B) / 2, (snd A + snd B) / 2)"
> and h2 : "M2 = ((fst B + fst C) / 2, (snd B + snd C) / 2)"
> and h3 : "M3 = ((fst A + fst C) / 2, (snd A + snd C) / 2)"
> and h4 : "S = fst M1 + fst M2 + fst M3"
> shows "S powr 2 = 13"
>
> theorem
> fixes A B C M1 M2 M3 :: "real * real"
> assumes h0 : "fst A + fst B + fst C = sqrt 13"
> and h1 : "M1 = midpoint A B"
> and h2 : "M2 = midpoint B C"
> and h3 : "M3 = midpoint C A"
> and h4 : "\<forall> P Q. midpoint P Q = ((fst P + fst Q)/2, (snd P + snd Q)/2)"
> and h5 : "S = fst M1 + fst M2 + fst M3"
> shows "S^2 = 13"
>
> (* What is the least common multiple of all positive integers smaller than 8? *)
>
> theorem
> fixes n :: nat and lcm :: "nat \<Rightarrow> nat \<Rightarrow> nat"
> assumes h0 : "n < 8"
> and h1 : "\<forall> a b. lcm a b = (a * b) div (gcd a b)"
> shows "fold lcm [1..<8] 1 = 420"
>
> theorem
> fixes lcm :: "nat \<Rightarrow> nat \<Rightarrow> nat"
> assumes h0 : "\<forall> a b. lcm a b = (a * b) div (gcd a b)"
> shows "lcm (lcm (lcm (lcm (lcm (lcm (lcm 1 1) 2) 3) 4) 5) 6) 7 = 420"
>
>
> (* Daniel works at an electronics store, and he claims that the popularity of a television (measured in number of sales) is inversely proportional to its cost.
> If 15 customers buy a television that costs $\\$$1500, according to Daniel's theory, how many customers would buy a television that costs $\\$$2500? *)
>
> theorem
> fixes cost1 cost2 sales1 sales2 :: real
> assumes h0 : "cost1 = 1500"
> and h1 : "cost2 = 2500"
> and h2 : "sales1 = 15"
> and h3 : "cost1 * sales1 = cost2 * sales2"
> shows "sales2 = 9"
>
> theorem
> fixes x y k :: real and f :: "real \<Rightarrow> real"
> assumes h0 : "f 1500 = 15"
> and h1 : "f 2500 = y"
> and h2 : "\<forall> x. x * f x = k"
> shows "y = 9"
> ```

---

> > ### Comment · Reviewer_LZ2a · 2024-08-12
> >
> > These examples are illuminating and help me better understand that the symbolic equivalence is checking for more than just variable name differences. Thanks to the authors for the responses, which have alleviated my previous concerns. Consequently, I will increase my score.

---

### Official Review · Reviewer_zk1U · 2024-07-11

**Soundness:** 4
**Presentation:** 4
**Contribution:** 4
**Rating:** 8
**Confidence:** 4

**Summary:**

This paper introduces two methods metrics, symbolic equivalence and semantic consistency, to determine the equivalence relation between formal statements. They design novel technical solutions to measure each method and show that this improves autoformalization by allowing self-consistency-based clustering.

**Strengths:**

1. The paper addresses an important and novel problem of improving autoformalization with more test-time compute.
2. The methods used are very intuitive and easy to understand.
3. The evaluation is sound and shows good performance gain.
4. The writing is clear and easy to follow.

**Weaknesses:**

Since the authors investigated 5 models of different sizes, I wish they investigated the models' performances with the same amount of test-time compute. I.e., given the same amount of FLOP (as measured by cost, since Codex and GPT-4 did not reveal their parameter count), which model is preferred. This helps the reader understand which model they should choose for their workload.

**Questions:**

How are the numbers supporting Figure 2 obtained? How many datapoints from MATH and miniF2F did you evaluate?

**Limitations:**

In the conclusions, the authors did mention the things they wish to leave for future work, but it feels somewhat vague. It would be good if the authors can be more concrete and lay out future ideas they'd like to try or would like the community to carry out.

---

> ### Author Rebuttal · Authors · 2024-08-07
>
> **Dear Reviewer zk1U**
>
> Thank you for the insightful feedback on our paper. We appreciate the time and effort you have put into reviewing our work, and we are grateful for the recognition of the importance of our research problem and encouraging comments such as clear writing, easy-to-understand, and good performance. We have carefully read your review and addressed your concerns as follows.
>
> **[Question #1] Autoformalization cost of different LLMs**
>
> Thank you for the suggestion. To assess the autoformalization cost of various LLMs, we calculate the number of output tokens generated per problem and present the results in the following. Considering the pricing for LLM generation, CodeX offers superior cost-effectiveness, since it not only achieves comparable performance with GPT-4 but also incurs a cost that is less than one-fifth of GPT-4's expense. We will include these results in the revision.
>
> | MATH | #Tokens | 1@k | miniF2F | #Tokens | 1@k |
> | --- | --- | --- | --- | --- | --- |
> | Mistral-7B | 2234.7 | 42.6 | Mistral-7B | 2049.7 | 18.0 |
> | Llemma-34B | 2036.9 | 45.2 | Llemma-34B | 1660.5 | 35.0 |
> | DeepSeek-v2 | 1848.9 | 46.9 | DeepSeek-v2 | 1698.1 | 29.9 |
> | CodeX | 1655.9 | 49.4 | CodeX | 1475.8 | 39.1 |
> | GPT-4 | 1446.4 | 45.3 | GPT-4 | 1360.3 | 43.6 |
>
> **[Question #2] Data sizes used in Figure 1 and evaluation**
>
> For the MATH dataset, we randomly selected a subset of 400 problems. As to the miniF2F dataset, we use the whole 488 problems. Figure 1 is obtained under the same setting. More details regarding the experimental setup are included in Section 4.1.
>
> **[Limitation] Vague future work**
>
> We appreciate your suggestion. The future work along with the proposed framework may lie in (1) adapting the proposed framework to the LEAN prover; (2) incorporating better large language models such as ProofGPT [1] and MMA [2] to further improve the performance of the framework; (3) generating more high-quality aligned informal and formal data using our framework; (4) generalizing our framework to proof autoformalization. We will update our paper to further specify these future directions.
>
> [1] ProofNet: Autoformalizing and Formally Proving Undergraduate-Level Mathematics
>
> [2] Multilingual Mathematical Autoformalization

---

> > ### Comment · Reviewer_zk1U · 2024-08-09
> >
> > Thank you for the response. They have addressed my question. I'll maintain my score.

---

### Official Review · Reviewer_37Eh · 2024-07-14

**Soundness:** 3
**Presentation:** 3
**Contribution:** 3
**Rating:** 7
**Confidence:** 2

**Summary:**

The paper presents a novel framework aimed at enhancing the accuracy of autoformalization in mathematics using LLMs. It addresses the challenge of translating natural language mathematical statements into formal language by introducing a two-pronged approach based on symbolic equivalence and semantic consistency. Symbolic equivalence leverages automated theorem provers to ensure logical homogeneity among autoformalization candidates, while semantic consistency evaluates the preservation of the original statement's meaning by comparing embeddings of the re-informalized text and the original. The framework scores and selects the best result from multiple autoformalization candidates, significantly improving accuracy across various LLMs and datasets like MATH and miniF2F. The extensive experiments demonstrate the synergistic effect of the two consistency methods, achieving relative improvements in autoformalization accuracy and reducing the need for manual verification.

**Strengths:**

1. Clear Presentation and Structure: The paper is well-organized, with a nice logical flow. I really like the figures and tables in the paper.
2. Open Source: The authors submit the implementation of the proposed methods, which is beneficial for the research community.
3. Innovative Framework: Introduces a new approach that combines symbolic equivalence and semantic consistency to improve the accuracy of autoformalization. Symbolic equivalence and semantic consistency are mutually complementary, with their combination further enhancing performance.
4. Comprehensive Evaluation: The authors conduct extensive experiments on two mathematical datasets, MATH and miniF2F, demonstrating the efficacy of the proposed methods.
5. Wide Applicability: The proposed framework is effective across various model sizes and is not limited to a specific LLM, indicating broad applicability.

**Weaknesses:**

This paper makes a valuable contribution to the field of formal mathematics. While I'm new to autoformalization and not an expert in this area, I believe the paper is well-written and presents a strong argument. However, due to my limited experience, my confidence level is relatively low as I might have missed some key points.

**Questions:**

I only have a few questions as below.
1. According to Table 1, It seems to me that semantic consistency is not working alone. It has to be used with Symbolic Equivalence. Could you provide more insights into why this is the case? On the other hand, the default settings of semantic consistency use representation from BERT and cosine similarity. Did you try other alternative methods to compute semantic consistency? I am not sure if the bottleneck is the idea of semantic consistency or the method used.
2. I am wondering if the proposed method also works for "low-resource autoformalization". The community is switching from Lean 3 to Lean 4, but existing LLMs are not performing well on Lean 4 as they have not been pretrained on enough Lean 4 corpus (also because of the limited availability).

**Limitations:**

The authors adequately addressed the limitations

---

> ### Author Rebuttal · Authors · 2024-08-07
>
> **Dear Reviewer 37Eh:**
>
> Thank you for the insightful feedback on our paper. We appreciate the time and effort you have put into reviewing our work, and we are grateful for encouraging comments such as clear presentation, comprehensive evaluation, and wide applicability. We have carefully read your review and addressed your concerns as follows.
>
> **[Question #1.1] Illustration of the semantic consistency**
>
> The performance of semantic consistency relies on the informalization process, which is relatively sensitive to the category and difficulty of the problem (see Table 2 and Table 4). While semantic consistency can work well for certain problems, this sensitivity can lead to inconsistent performance when used alone. On the other hand, symbolic equivalence provides a more stable performance by computing the score based on the size of the derived equivalence class, as seen in Figure 6. When combined, semantic consistency and symbolic equivalence complement each other effectively. Semantic consistency helps identify the optimal version among multiple large symbolic equivalence classes, thereby enhancing overall performance.
>
> **[Question #1.2] Alternative methods to compute semantic consistency**
>
> We appreciate your suggestion to explore alternative methods. As suggested, we use e5-Mistral [1] as an alternative to BERT for our embedding model. The results are detailed below. We can observe that the new embedding model does not make a significant impact on the results. This observation suggests that the quality of informalization is more critical for semantic consistency than the specific embedding model used.
>
> | MATH |  |  |  | miniF2F |  |  |  |
> | --- | --- | --- | --- | --- | --- | --- | --- |
> | SemCo(BERT) | SemCo(e5-Mistral) | Log-comb (BERT) | Log-comb (e5-Mistral) | SemCo(BERT) | SemCo(e5-Mistral) | Log-comb (BERT) | Log-comb (e5-Mistral) |
> | 39.5 | 40.5 | 45.3 | 46.7 | 34.9 | 34.4 | 43.6 | 42.6 |
>
> **[Question #2] Low-resource autoformalization**
>
> Although we have not yet adapted our framework to LEAN (this is planned for future work). However, the results from Table 1 and Figure 5 demonstrate that our framework remains effective even with models exhibiting low autoformalization performance (e.g., a significant improvement from 18.1% to 42.6% and from 7.5% to 18.0% for Mistral 7B on MATH and MiniF2F, respectively). Therefore, the efficacy of our framework could be expected even if the low-resource degrades the model performance.
>
> [1] https://huggingface.co/intfloat/e5-mistral-7b-instruct

---

> > ### Comment · Reviewer_37Eh · 2024-08-09
> >
> > Thanks for the response. The authors have addressed most of my concerns. I will keep my rating positive for t his paper.

---

### Author Rebuttal · Authors · 2024-08-07

We thank all the reviewers for their valuable and insightful comments, which help improve our paper accordingly. Here, we summarize our responses to the major issues raised by the reviewers.

Reviewer **37Eh** and **LZ2a** request a further discussion about the performance of semantic consistency. First, we have experimented with a more state-of-the-art embedding model (i.e., e5-Mistral [1]) and find the limited effect of using different embedding models (detailed results in the response to reviewer **37Eh** and **LZ2a)**. Moreover, as indicated by Table 2 and Table 4, it appears that the performance of semantic consistency is more affected by the performance of informalization.

Reviewer **LZ2a** and **GLbW** suggest further clarification on the technical details of semantic consistency and symbolic equivalence. First, we clarify more details about the semantic consistency, including the combination strategy, and the definition of $\tau$-semantic consistency (details in the response to reviewer **LZ2a**). We also have detailed examples to explain the rationale and the benefit behind the definition of symbolic equivalence (details in the response to reviewer  **GLbW**).

Reviewer **37Eh** and **GLbW** request an analysis of our framework on other settings. First, the results of Mistral 7B from Table 1 and Figure 5 indicate the effectiveness of our methods even when the model achieves low autoformalization performance (details in the response to reviewer  **37Eh**). We also included an experiment on the Mistral 7B model fine-tuned on the MMA dataset, confirming the efficacy of the proposed framework (detailed results in the response to reviewer **GLbW**).

Reviewer **zk1U** requests a cost comparison on different models. Based on the number of output tokens generated per problem with different models, it suggests that CodeX offers superior cost-effectiveness (results in the response to reviewer **zk1U**)

Reviewer **zk1U** and **GLbW** figure out typos in the text and Figure 1. We really appreciate their efforts and will revise our paper accordingly with more careful proofreading.

---

### Decision · Program_Chairs · 2024-09-25

**Decision:**

Accept (poster)

**Comment:**

This paper presents a method for improving autoformalization of natural language statements through a modification of self-consistency. Specifically, the paper uses symbolic equivalence checking to marginalize over different formalizations of the statement like standard answer match in self-consistency. Semantic consistency then examines whether a re-informalization of these statements is equivalent with the original. Results show that symbolic equivalence checking is very helpful and a log combination with semantic consistency helps a bit on MATH and miniF2F.

The reviewers generally found the paper to present a nice idea with reasonably strong results. It's well-written.  One weakness is the fact that semantic consistency doesn't work all that well and parts of its implementation are a bit ad hoc. Moreover, the symbolic equivalence checking is a natural extension of the well-studied self-consistency method to this setting.  Nevertheless, the paper does a good job fleshing this out and conducting experiments with it.